# Pleiotropic Action of TGF-Beta in Physiological and Pathological Liver Conditions

**DOI:** 10.3390/biomedicines12040925

**Published:** 2024-04-22

**Authors:** Michał Jakub Braczkowski, Klaudia Maria Kufel, Julia Kulińska, Daniel Łukasz Czyż, Aleksander Dittmann, Michał Wiertelak, Marcin Sławomir Młodzik, Ryszard Braczkowski, Dariusz Soszyński

**Affiliations:** 1Department of Physiology, Institute of Medical Sciences, University of Opole, 45040 Opole, Poland; dsoszynski@uni.opole.pl; 2Student Scientific Society of Physiology, Department of Physiology, Institute of Medical Sciences, University of Opole, 45040 Opole, Poland; klaudiakufel@interia.pl (K.M.K.); kulinskajulia@gmail.com (J.K.); aleksander.dittmann@uni.opole.pl (A.D.); michal.wiertelak.mw@gmail.com (M.W.); 3Department of Pathology, Institute of Medical Sciences, University of Opole, 45040 Opole, Poland; marcin.mlodzik@uni.opole.pl; 4Specialist Hospital No. 1, 41900 Bytom, Poland; rbracz@interia.pl; 5Department of Human Physiology, Faculty of Medicine, Collegium Medicum in Bydgoszcz, Nicolaus Copernicus University in Torun, 87100 Torun, Poland

**Keywords:** TGF-β, pleiotropism, liver diseases, apoptosis, liver fibrosis, liver inflammation, liver steatosis

## Abstract

The aim of this study is to review and analyze the pleiotropic effects of TGF-β in physiological and pathological conditions of the liver, with particular emphasis on its role in immune suppression, wound healing, regulation of cell growth and differentiation, and liver cell apoptosis. A literature review was conducted, including 52 studies, comprising review articles, in vitro and in vivo studies, and meta-analyses. Only studies published in peer-reviewed scientific journals were included in the analysis. TGF-β is a pleiotropic growth factor that is crucial for the liver, both in physiology and pathophysiology. Although its functions are complex and diverse, TGF-β plays a constant role in immune suppression, wound healing, and the regulation of cell growth and differentiation. In concentrations exceeding the norm, it can induce the apoptosis of liver cells. Increased TGF-β levels are observed in many liver diseases, such as fibrosis, inflammation, and steatosis. TGF-β has been shown to play a key role in many physiological and pathological processes of the liver, and its concentration may be a potential diagnostic and prognostic marker in liver diseases.

## 1. Introduction

Transforming growth factor TGF-β is a prototype of a large and constantly growing superfamily. This family includes over forty structurally related multifunctional protein factors found in vertebrates, *Caenorhabditis elegans*, and *Drosophila melanogaster* [1]. Individual members of the TGF-β family are synthesized and secreted by cells as large propeptides, which, upon proteolytic activation, affect cell growth and differentiation, the body’s immune response, the organization of the extracellular matrix (ECM), and biogenesis [1,2].

The TGF-β subfamily includes five isoforms, of which TGF-1, TGF-2, and TGF-3 are found in mammals, isoform TGF-4 in chickens, and TGF-5 in the frog Xenopus laevis [1,3]. The first member of the TGF-β subfamily was described in 1981 as an element induced by epidermal growth factor (EGF) and capable of stimulating the transformation of fibroblasts into a tumor phenotype. However, later observations showed that TGF-β is a factor that inhibits tumor development to a greater extent than it stimulates it [4]. Depending on the cell type, TGF-β can stimulate or inhibit growth. Isoforms 1, 2, and 3 show similar activity in vitro, but in vivo, they are expressed to varying degrees and have different functions [3].

In addition to the transforming factor mentioned above, the TGF-β superfamily in vertebrates includes bone morphogenetic proteins (BMPs), activin, and inhibin [3]. Related to them are the DPP gene (Decapentaplegic gene) of *Drosophila melanogaster* and the DAF and DBL genes of *Caernorhabditis elegans* [5].

Activin and inhibin are protein hormones produced in ovarian follicles; they are characterized by opposite effects in the body:Activin stimulates the secretion of FSH by the pituitary gland; inhibin inhibits the secretion of FSH;Activin stimulates immune response processes (increased gamma interferon and activation of monocyte migration); inhibin inhibits immune processes (decreased gamma interferon);Activin stimulates erythropoiesis, inhibin inhibits it;In addition, activin mediates the stimulation of the dorsal mesoderm in embryonic development [6].

Bone morphogenetic proteins (BMPs) are the largest subgroup of the TGF-β superfamily, comprising over 30 members [7]. They occur as natural factors that play an important role in the processes of bone formation during embryogenesis as well as its remodeling in adulthood. Additionally, they play an important role during morphogenesis; during chondrogenesis, BMP 2 and 4 were found to be expressed in the primary, undifferentiated mesenchyme. In turn, the correct course of cartilage tissue differentiation takes place with the expression of BMP-2 and BMP-3 [8]. In embryogenesis, BMPs also participate in the differentiation of nervous tissue and in organogenesis. BMPs have been shown to inhibit the growth of smooth muscle cells, breast cancer cells, and bone cancer cells [6].

The secretion of TGF-β is induced by, among others, steroids, retinoids, epidermal growth factor (EGF), nerve growth factor (NGF), vitamin D3, and interleukin 1 (IL-1). TGF-β synthesis can be inhibited by EGF, fibroblast growth factor (FGF), dexamethasone, calcium, retinoids, and follicle-stimulating hormone (FSH) [9].

## 2. Structure, Synthesis, Receptors and Signaling Pathways

### 2.1. Structure and Synthesis of TGF-β

TGF-β is synthesized in cells as a homodimeric precursor of about 100 kDa, which undergoes intracellular proteolytic cleavage in the Golgi apparatus. The enzyme involved in the cleavage process is furin, which belongs to the class of endopeptidases [5,10]. Furin cleaves the precursor molecule at a specific RXXR site, located at a distance of 112–114 amino acids from the C-terminus of the molecule, into two fragments: the C-terminal fragment—the mature 25 kDa dimeric TGF-β—and the N-terminal 75 kDa fragment called the latency associated peptide (LAP) (Figure 1) [3,6,9].

LAP remains non-covalently bound to two cysteine residues at positions 223 and 225 in mature TGF-β, forming an inactive complex called the small latent complex (SLC) [3,6]. This “small latent” TGF-β complex is more stable than the bioactive TGF-β [6]. In the Golgi apparatus, disulfide bonds can also form between LAP and LTBP (latent TGF-binding protein) to form the so-called large latent complex (LLC), which is over 225 kDa in size [3,10]. LTBP exists as four isoforms, numbered 1 to 4. All four isoforms of LTBP have been localized in the human liver.

The role of LTBP is to increase the stability and mediate the secretion of inactive TGF-β complexes [6]. In addition, LTBP allows the latent TGF-β complex to bind to some ECM proteins, enabling the formation of a reservoir in the extracellular matrix from which active TGF-β can be released without the need for de novo synthesis [5]. Isoforms 1, 2, and 4 of LTBP contain RGD sequences, which are the binding sites for integrins. It has been shown that the latent TGF-β complex, after secretion from the cell, can directly interact with integrin v/1 on the cell surface, which can activate integrin signaling [6].

TGF-β is therefore released from cells as inactive complexes (SLC or LLC), which are unable to bind to type I, II, and III transmembrane receptors (TRI, TRII, and TRIII) [6]. Biological activity is possessed only by free TGF-β, which must be released from the inactive complex by dislodging LAP [4]. In vivo, activation of the latent TGF-β complex involves binding of LAP to mannose-6-phosphate receptors and proteolytic cleavage of LAP by plasmin and cathepsin [3,10,11]. Another important in vivo activator of the TGF-β complex is thrombospondin-1, which causes conformational changes in the LAP structure, thereby activating TGF-β [1,3,6]. In vitro, the inactive TGF-β complex can be activated by treatment with acids, bases, urea, heat, and proteases [12].

Optimal activation of TGF-β requires the presence of extracellular matrix (ECM) components associated with fibrosis processes (including collagens, fibronectin, laminin, and tenascin), as well as interaction with pericytes and CD8+ lymphocytes. Recently, the existence of a LTBP-1D variant has been confirmed that is less sensitive to the action of proteolytic enzymes, which makes it difficult to activate TGF-β. The half-life of the latent TGF-β complex in circulation is 90 min, and that of the free form is about 3 min, after which the molecule is eliminated by binding to 2-microglobulin. However, this time is relatively long compared to other cytokines, which allows for measurements of the concentration in circulation [4]. According to the authors of other publications, 2-macroglobulin is involved in the binding and then elimination of TGF-β from the bloodstream [1,10].

### 2.2. Receptors and Signaling Pathways

The signaling activity of TGF-β towards a cell depends on the presence of specific membrane receptors, which are present as dimeric proteins with molecular masses of 75 and 53 kDa [4]. The intracellular fragments of receptors I and II have serine–threonine kinase activity domains [8]. TβRII can bind the growth factor in question independently of the presence of TβRI but is not capable of transducing the biological signal [2]. In the absence of the type II receptor, cells are insensitive to TGF-β [13]. The extracellular part of TRII binds the mature ligand (TGF-β), activating the intracellular domain of the receptor. The resulting complex binds the type I receptor, which determines the specificity of ligand recognition [4]. The activated type II receptor kinase phosphorylates the serine residues of the TTSGSGSG sequence in the GS domain (glycine- and serine-rich domain) in the type I receptor, thereby activating the serine-threonine kinase in that receptor, leading to the initiation of a signal transduction cascade into the cell interior (Figure 2) [3,5].

TGF-β can also bind to the type III receptor (TβRIII)—the so-called betaglycan, which is a transmembrane proteoglycan that does not exert any intracellular action; its role is to present TGF-β to the other two receptors [8]. TβRIII, with a molecular weight of >250 kDa, is the most abundant receptor subtype in most cells [14,15]. It shows high affinity for all three isoforms of TGF-β found in mammals but is a receptor required for the binding of the TGF-2 isoform to the type II receptor. In its absence, cells do not respond to TGF-II, while the response of cells to the other two isoforms of the cytokine is preserved [3,5].

Another auxiliary receptor is endoglin (structurally similar to betaglycan). Endoglin, being a transmembrane glycoprotein, shows high affinity for the isoforms TGF-1 and TGF-3 but does not bind TGF-2 [5]. Endoglin, being strongly preferentially expressed in endothelial cells of blood vessels, plays a crucial role in angiogenesis and tumor progression [16].

After the activation of serine-threonine kinase in TβRI, the signal is further transmitted by phosphorylation of the cytoplasmic proteins Smad-2 and Smad-3 that bind to the receptors. Smad 2 and 3 proteins belong to the so-called R-Smad group (receptor-activated Smads–R-Smad). The phosphorylated R-Smad proteins form a complex with the so-called co-Smad (common partner Smad), i.e., Smad-4; the resulting complex is transported to the cell nucleus [10,17]. Signal transduction from the receptor to the nucleus can be inhibited by Smad-6 and Smad-7 (so-called I-Smads, i.e., inhibitory Smads), which, despite the lack of a motif in their structure that can be phosphorylated by receptor kinases, are able to interact with membrane receptors, thereby impairing their interaction with R-Smad [7,17]. The expression of Smad-7 protein is induced by TGF-β, which leads to the inhibition of the cellular response to the cytokine in question (negative feedback mechanism) [10,17]. Transcription of the gene encoding Smad-7 is also stimulated by interferon gamma (IFN-γ), which explains the antifibrogenic effect of IFN-γ [5]. The auxiliary protein SARA (Smad anchor for receptor activation) is involved in the early stages of signal transduction. SARA is a cytoplasmic protein anchored in the cell membrane, binding both R-Smad (2 and 3) and the TGF-/TβRII/TβRI complex. The SARA protein, recognizing the unphosphorylated R-Smad, binds it to the formed receptor complex and itself dissociates [6,10].

Cytosolic Smad proteins 2, 3, and 4 mediate signal transduction from the receptors with which TGF-β binds to the cell nucleus. However, neither Smad-2 nor Smad-4 have the ability to bind nuclear DNA, while Smad-3 binds weakly to nucleic acids. Therefore, in the cell nucleus, Smad proteins modulate the transcription of individual genes after forming a complex with transcription factors, or so-called co-repressors or co-activators of transcription. The group of transcription co-activators includes c-jun and lef1, while the co-repressors include SNIP1, Sno-N, and Ski. In the case of Sno-N, a two-phase change in the amount of this co-repressor was observed. Immediately after the signal from TRI reaches the nucleus, there is a rapid degradation of Sno-N, which allows transcription to occur. After some time, Sno-N is resynthesized, and its amount increases. The resynthesis of Sno-N indicates that the gene encoding Sno-N belongs to the genes transcriptionally activated by TGF-β. Moreover, the resynthesis of the co-repressor limits the duration of the signal coming from the activated membrane receptors [5].

At the AP-1 promoter site for human collagenase 1 (MMP-1), there is an interaction between c-Jun/c-Fos and Smad-3 and Smad-4, which results in the induction of transcription. In the PAI-1 gene, the Smad-3/Smad-4 complex cooperates with the transcription factor TFE3 to regulate transcription. Newly discovered regulatory elements may also play an important role in the process of fibrogenesis: the induction of a 30 kDa nuclear protein and its binding to the recently described promoter element during the activation of stellate cells play an important role in stimulating the expression of tissue inhibitor of metalloproteinases 1 (TIMP-1) [17].

### 2.3. Alternate Pathways of Action

Another cytosolic protein that affects signal transduction from activated TRI to the nucleus is FKBP12. FKBP12 is a protein that binds rapamycin and tacrolimus (FK506) and, in the absence of these factors, binds to TRI near its GS domain, thereby blocking phosphorylation of that domain by the type II receptor kinase. The expected consequence of delivering rapamycin or tacrolimus to the cell is therefore an increase in signals from the membrane receptors of the transforming growth factor beta.

It remains unknown whether FKBP12 is present in hepatic stellate cells; however, it is known that FK506 accelerates the process of liver fibrosis in rats after experimental carbon tetrachloride damage. The question of whether administration of FK506 to patients with liver damage will increase the process of liver fibrosis in them has not been answered either [5].

Mutations located in the type I receptor between the transmembrane domain and the GS domain are responsible for the elimination of some, but not all, of the effects of TGF-β on target cells. For example, serine at position 172 and threonine at position 176 are essential for the growth inhibitory effect of TGF but do not affect the increased production of fibronectin and PAI-1 (plasminogen activator inhibitor-1). These amino acids are not phosphorylated; however, the mechanism by which they modulate signal transduction is not known. Similarly, in the type II receptor, Thr at position 315 (not autophosphorylated) is required for cell growth inhibition but does not affect extracellular matrix induction [17].

## 3. Biological Significance of TGF-β

### 3.1. Biological Activity

Transforming growth factor beta (TGF-β) is an endogenous polypeptide substance that has its own cellular receptors. Like other cytokines, it plays a key role in intercellular communication. It is a molecule involved in phenomena occurring in the body, both in health and disease [4,11].

The natural source of TGF-β is platelets, from which the cytokine is released immediately as a result of their degranulation and occurs directly after injury or as a result of immunological reactions. In addition, exceptionally high concentrations of TGF-β are also found in bone tissue, the spleen, and the placenta. The concentration of this cytokine in bone tissue is about 100 times higher than in other tissues and is about 200 mg/kg of tissue [4,8,15,18]. In vitro, TGF-β isoforms exert a similar biological effect on tissues. However, in vivo, they are characterized by a fundamentally different degree of expression and perform different functions. The dominant form of transforming growth factor in the human body is TGF-1, synthesized by almost all cells, and its receptors are found on the surface of most of them. The remaining isoforms are expressed in a limited range of cells and tissues. The strongest expression of TGF-3 is observed in the embryonic heart and lung tissue, while the weakest production of this isoform occurs in the liver, spleen, and kidneys. TGF-2, on the other hand, is synthesized in significant amounts in glioblastoma cells and keratinocytes [4].

The consequence of the spread of TGF-β in the body is its significant impact on most physiological processes. The resulting interactions with other cytokines and biologically active substances, as well as organ and tissue specificities, make it difficult to unequivocally characterize TGF-β. However, there are certain constant functions of this cytokine that are not controversial.

TGF-β is involved in fetal development, growth regulation and cell differentiation, tissue modification, wound healing, immune suppression, tumor growth inhibition, and extracellular matrix composition regulation. The described cytokine is also involved in the pathogenesis of fibrosis, immunosuppression, tumor development, autoimmunity, inflammatory diseases, and angiogenesis [4].

The effect of TGF-β on fetal development is reflected in the developmental defects observed in transgenic mouse models with deletion of the gene for individual isoforms of this cytokine. Half of the mice lacking the gene encoding the TGF-1 isoform die in utero due to abnormal vasculogenesis and hematopoiesis. The remaining TGF-1 knock-out mice are born without visible developmental defects but die within three weeks of birth. The cause of their deaths is multifocal inflammatory states of the liver and other tissues [5,6,17,18]. TGF-2-null mice die before birth due to severe and multiple developmental defects in many tissues and organs, especially the heart, lungs, limbs, and inner ear. In turn, transgenic TGF-3 knock-out mice exhibit delayed lung development and a cleft palate and die shortly after birth [1,6]. TGF-β is a pluripotent growth factor that stimulates cells of mesenchymal origin and inhibits cells of neuroectodermal origin [8].

TGF-β stimulates the growth of some types of cells of mesenchymal origin. An example is the stimulating effect on fibroblasts and osteoblasts, both in vitro and in vivo. On the other hand, isoforms TGF-1 and -2 enhance the proliferation of Schwann cells [9].

TGF-β also exerts a strong direct and indirect systemic pro-inflammatory and immunosuppressive effect [11].

### 3.2. Pathogenic Effects of TGF-β

TGF-β inhibits DNA synthesis in many cells, which impairs their growth. It does this by blocking the transition of cells from the G1 phase to the S phase of the cell cycle. TGF-β is the strongest known inhibitor of the growth of normal and transformed epithelial cells, endothelial cells, fibroblasts, nerve cells, lymphatic cells, hematopoietic cells, hepatocytes, and keratinocytes. Only the TGF-2 isoform does not inhibit the growth of endothelial cells [19].

The extent of growth inhibition by TGF-β depends on the cell type, the concentration of the cytokine, and the interaction of other biologically active substances. For example, lung epithelial cells and keratinocytes are the most sensitive to the growth-inhibitory effects of TGF-β. At a concentration of 1–2 fg/cell, TGF-β inhibits the growth of smooth muscle cells, fibroblasts, and chondrocytes. At higher concentrations, the cytokine has the opposite effect—it stimulates the growth of these cell types. Hepatic growth factor (HGF) has the ability to abolish the inhibitory effect of TGF-β.

The discussed growth factor plays an important role in the inflammatory response, as it regulates the production of acute phase proteins and decreases the synthesis of albumin and fibrinogen. It is therefore an important modulator of inflammation and tissue repair. It also increases the secretion of alpha 1 antichymotrypsin and decreases the synthesis of alpha-fetoprotein. However, it does not affect the synthesis of haptoglobin [19].

The suppressive effect of TGF-1 on the immune system increases the body’s susceptibility to infections. This cytokine inhibits the proliferation of B and T lymphocytes and the cytotoxic activity of NK cells. It also has an inhibitory effect on the maturation process of macrophages and blocks the production of cytokines. TGF-β stimulates the secretion of immunoglobulin class A (IgA) by B lymphocytes, while the secretion of immunoglobulin classes G and M (IgG and IgM) is inhibited by it. TGF-β administered chronically to the circulation inhibits the expression of selectins and receptors for interleukin-1 (IL-1), thereby lowering adhesion [9,11,18].

Among the many functions of TGF-β, one of the most important is its ability to stimulate the synthesis of extracellular matrix proteins, which is essential for wound healing and tissue remodeling [4].

Confirmation of the pro-inflammatory and profibrogenic activity of TGF-1 in vivo was provided by the results obtained in a transgenic mouse model generated by microinjection of the fusion gene Alb/TGF-1 into single-cell mouse embryos. In the described model, there was selective expression of the mature form of TGF-1 in hepatocytes (due to the replacement of cysteine at positions 223 and 225 of the TGF-β propeptide with serine). The hepatic overexpression of TGF-1 was reflected by an approximately 10-fold increase in the level of this cytokine in the plasma of transgenic animals compared to the control group. The most intense degree of liver fibrosis was observed in 10–12-week old mice, which was accompanied by a high level of expression of the gene encoding type I collagen in this organ. The elevated level of TGF-1 in the plasma, especially in the first 6 weeks of life, caused extrahepatic pathological changes such as inflammatory and fibrotic changes in the kidneys, inflammatory changes in the heart and small and medium-sized cardiac and renal arteries, and atrophic changes in the pancreas [12].

The most attention is paid to the role of TGF-β in neoplastic diseases, where the lack of TGF-1-dependent control of cell growth may be responsible for oncogenesis. An example is the observed lack of the suppressor gene DPC4 in pancreatic cancer, which is responsible for the synthesis of the protein Smad-4, which is an effector of TGF-1. In turn, defects in the expression of the type II receptor for TGF-β have been described in gastric cancer and malignant lymphoma. In cells derived from colon cancer, a mutation was found that causes the lack of activity of the gene responsible for the synthesis of TβRII. The introduction of copies of this gene into the cells led to the reversal of oncogenesis [4].

Under normal conditions, TGF-β is a potent inhibitor of the growth of many cell types, including cancer cells. However, once cancer cells enter a phase of uncontrolled growth, most of them lose their sensitivity to the growth-inhibitory effects of TGF-β. Surprisingly, this occurs despite the presence of receptors for this cytokine on the surface of cancer cells. Additionally, these cells themselves begin to secrete the indicated cytokine [20].

An example is liver cancer cells, which produce active TGF-β, in contrast to normal and chemically transformed liver cells, which produce an inactive form of this growth factor [19]. Some researchers hypothesize that under conditions of uncontrolled growth of cancer cells, TGF-β plays a different role than previously known, which is not yet fully understood [20]. Animal studies have shown that TGF-β increases the invasiveness of cancer cells [19]. Overproduction of TGF-1 by cancer cells may contribute to neovascularization and suppression of the immune response, thus facilitating tumor development in vivo [9].

### 3.3. TGF-β and Apoptosis

TGF-β at a concentration higher than the growth inhibitory concentration induces apoptosis of liver cells in culture. Liver parenchymal cells—hepatocytes—are more sensitive to the apoptotic effect of TGF-β than non-parenchymal cells, such as stellate cells [5,10]. It turned out that the apoptosis process in the primary rat hepatocyte line and in the human hepatoma cell line is associated with the suppression of phosphorylation of the retinoblastoma gene product pRb.

The Rb gene product (Rb protein) is located in the cell nucleus and is one of the main inhibitors of DNA replication in the cell cycle. The Rb protein exists in two forms: phosphorylated and non-phosphorylated. In its non-phosphorylated form, it binds to certain gene regulatory proteins and prevents them from activating the DNA replication process. During TGF-β-induced apoptosis in cultured rat hepatocytes, an increase in the activity of caspases 3 and 8 was observed [10]. Similarly, studies conducted in the human hepatoma (hepatocellular carcinoma) cell line Huh7 showed that part of the early response to TGF-β was an increase in the activity of caspase-8, followed by the activation of caspase-9 and caspase-3. On the other hand, caspase-8 inhibitors blocked the apoptosis process in the human hepatoma cell line. Interleukin 6 also prevented the activation of caspase-3, thus blocking apoptosis induced by the discussed cytokine [5].

Apoptosis of parenchymal cells is also associated with an increase in the concentration of reactive oxygen species (ROS) inside the cell and a decrease in the level of reduced glutathione. In the rat hepatoma cell line, the induction of cytosolic transglutaminase was directly associated with TGF-1-induced apoptosis [10]. TGF-1-induced apoptosis in hepatocyte culture can be effectively blocked by dexamethasone, phenobarbital, bacterial lipopolysaccharide, EGF, nuclear transcription factor kB (NFkB), and other apoptosis inhibitors [10].

## 4. The Importance of TGF-β in the Liver

### 4.1. Production of TGF-β by the Liver

TGF-β is secreted by normal liver non-parenchymal cells (macrophages and lymphocytes), platelets, chemically transformed cells, and liver cancer cells. Normal and transformed cells produce an inactive form of TGF-β, which is converted to the active form by cleavage of the carrier protein. On the other hand, cancer cells produce active TGF-β.

Although TGF-β affects hepatocytes in healthy and regenerating livers, these cells do not show the ability to produce it [19].

Presumably, however, TGF-β can be taken up by hepatocytes and released when their cell membrane is damaged [5,10]. The gene and mRNA for TGF-β are present in non-parenchymal liver cells [19]. A relatively high constitutive (unchanged in case of organ damage) level of mRNA for TGF-β1 is present in the liver in sinusoidal cells, endothelial (lining of blood vessels) and Kupffer cells, and epithelial cells of the bile ducts.

The mRNA levels for the other two isoforms of the cytokine in these cell types are much lower. On the other hand, hepatic stellate cells are characterized by a low level of TGF-β1 gene expression, which increases dramatically in the damaged organ. For example, in rat models after bile duct ligation, a linear increase in TGF-β1 expression in stellate cells was observed in the period of 12 h 7 days after ligation [5,21].

In the liver, TGF-β strongly suppresses hepatocyte proliferation, stimulates the production of extracellular matrix proteins by stellate cells, and mediates apoptosis. Unbalanced TGF-β activity during regeneration can lead to liver fibrosis [13].

### 4.2. TGF-β Involvement in Intercellular Signaling in the Liver

Studies involving the delivery of medium from Kupffer cell cultures to Ito cell cultures have indicated the possibility of paracrine signaling within the Disse space. Kupffer cells from animals previously exposed to carbon tetrachloride secrete a substance that enhances the proliferation and transformation of Ito cells. The increased synthesis of some cytokines by these cells is associated with their transformation into myofibroblasts. They affect the regeneration processes of liver parenchymal cells and the functions of other cells in their immediate vicinity [4].

Disruption of the integrity of liver cells under the influence of various factors causes the migration of mononuclear cells and non-parenchymal cells to the site of necrosis. Hepatocyte damage activates hepatic macrophages, Kupffer cells, and platelets. Phagocytosis of liver cells at the site of necrosis is accompanied by the release of cytokines: TGF-β1, TGF-α, and platelet-derived growth factor (PDGF). Activated macrophages can further damage hepatocytes by releasing proteases, accompanying cytokines (TNF-α), and superoxides, thus leading to a vicious circle. It is also possible that TGF-β1 can induce hepatocyte apoptosis. In the case of extensive liver parenchyma damage, the exposure of collagen fibers can be a direct stimulus for platelet adhesion and aggregation, which is further facilitated by the platelet-activating factor (PAF) released by Ito cells, which, in turn, release TGF-β1 and PDGF, which enhance the expression of Ito cells. While the action of TGF-β1 is to directly enhance the expression of genes responsible for the synthesis of ECM proteins, PDGF causes this effect indirectly by activating Ito cells (stellate cells). This mechanism involves the generation of a chemotactic gradient under the influence of PDGF, which conditions the migration of Ito cells. Their proliferation at the site of tissue damage is further enhanced by TGF-β1 derived from macrophages [4].

### 4.3. TGF-β Inhibition of Hepatocyte Growth

Under physiological conditions, hepatocytes are in the G0 phase and rarely divide. Acute and chronic liver inflammations (e.g., due to HBV infection), autoimmune diseases, and partial hepatectomy of the liver damage liver parenchymal cells and increase regeneration processes. As a result of the stimulation of liver regeneration activity, hepatocytes undergo 2–3 replication cycles, after which they return to the resting state.

Liver regeneration takes place in two phases. The role of initiating factors, which bring hepatocytes out of the G0 phase, is played by hormones (e.g., NA, insulin, glucagon, and ADH). Hepatocytes in the G1 stage are affected by two types of growth factors: full mitogens (e.g., TGF-α), which have the ability to activate nuclear DNA synthesis, and incomplete mitogens, which are compounds that only inhibit this process (e.g., TGF-β1). TGF-β is a strong antimitogenic factor for hepatocytes, and is probably the main inhibitor of uncontrolled hepatocyte proliferation. This factor inhibits the synthesis of nuclear DNA and the production of albumin and fibrinogen in hepatocytes and also reduces the activity of the strongest mitogen, HGF (hepatic growth factor) [22]. In addition to TGF-β, there are very few factors that inhibit the action of growth factors, such as glucocorticoids. In contrast, there are many more factors that promote hepatocyte proliferation. These include mainly HGF and TGF-α, but also IL-1β, TNF-α, insulin, growth hormone (GH), prostaglandin (PG), and serotonin (5-HT). This disproportion in the ratio of growth factors to inhibitory factors necessitates tight regulation between them. It has been demonstrated that even small amounts of TGF-β can strongly inhibit the hepatocyte proliferation-promoting effects of EGF, HGF, and TGF-α in culture [23]. 

The discussed cytokine turned out to be a strong inhibitor of hepatocyte growth in vitro and in the regenerating liver in a rat model after partial hepatectomy [5,10]. TGF-β inhibits hepatocyte proliferation by blocking their transition from the G1 to S phase (TGF-β induces cell cycle proteins p15 and p21) [5,19]. It is therefore the final element of the inhibitory paracrine chain, which is activated during liver regeneration. This mechanism probably prevents uncontrolled hepatocyte proliferation [19].

Norepinephrine modulates the hepatocyte response to TGF-β via the alpha receptor, increasing the sensitivity threshold to this factor. Prazosin, a drug that blocks the alpha 1 receptor, inhibits this effect of adrenaline. This effect could be important during liver regeneration, as it would allow hepatocytes to “escape” the negative control of TGF-β [19]. It has been reported in vivo that TGF-β1 increases in blood very rapidly within 2–3 h and is maintained for 72 h thereafter [24]. During the first half of liver regeneration (about 3 days), the inhibitory effect of TGF-β1 may not be visible due to the dominant effect of many growth factors. Only in the second half of liver regeneration (3 to 7 days), when the effect of many growth factors has ended, does the effect of TGF-β1 dominate, and liver regeneration is considered complete [23].

### 4.4. TGF-β Effect on Stellate Cells

Liver damage in experimental animal models, whether chemical with carbon tetrachloride or anatomical by bile duct ligation, is accompanied by a rapid increase in TGF-β expression, especially in stellate cells. In humans, repeated liver damage in chronic viral hepatitis, as well as damage caused by drug or alcohol abuse, leads to a significant increase in the synthesis of the discussed growth factor in hepatic stellate cells [13]. In patients with chronic liver diseases, a correlation is observed between the increase in serum TGF-β1 concentration and the severity of liver detoxification function impairment expressed by hyperbilirubinemia and the massiveness of hepatocyte damage manifested by aminotransferase activity. The highest TGF-β1 concentrations were observed in patients with autoimmune hepatitis (109 ng/mL) as well as with large liver hemangiomas (113 ng/mL). On the other hand, in patients with chronic hepatitis type C, there is no correlation between TGF-β1 concentration and hepatocyte damage [4].

Researchers tudied changes in TβRII and TGF-β1 expression caused by viral infection in patients with chronic HCV and bile duct ligation in a group of rats. The results showed that the mRNA level for TGF-β1 was significantly increased, while the expression of TβRII decreased both in biopsy samples of stellate cells from people with chronic HCV and in the group of rats after bile duct ligation. TGF-β expression in stellate cells in rats increased linearly from 12 h to 7 days after bile duct ligation. Five days after ligation, the mRNA level for TβRII in stellate cells in rats was 40% of the level observed in the control group of animals (in which the duct was not ligated). The TβRI receptor occurs in the rat liver in the form of two isoforms: TβR1 and Tsk7L, and it is believed that mainly the TβRI isoform is involved in TGF-β binding. After bile duct ligation, the decrease in mRNA for TβRII was accompanied by a significant increase in mRNA for the TβRI isoform in endothelial and stellate cells and an unchanged mRNA level for TskL7 [13]. Changes in the expression profile of TβRI and TβRII receptors correlated with two events in the damaged liver: increased ECM protein production and increased stellate cell proliferation. Stellate cell proliferation increased progressively in rats after bile duct ligation, suggesting a decreased sensitivity of stellate cells to the antiproliferative effect of TGF-β in the damaged liver. To confirm this observation, Researchers isolated liver stellate cells from rats with ligated bile ducts on day 5 after the procedure and from control rats and incubated them with 10 ng/mL TGF-1. After incubation, a significant decrease in DNA synthesis was observed in stellate cells isolated from control animals. On the other hand, stellate cells from rats with ligated bile ducts were found to be insensitive to the growth-inhibiting effect of TGF-1, and after incubation with the described cytokine, the decrease in DNA synthesis was significantly limited in these cells [13,21].

Later studies confirmed that the amount of TβRII actually decreases in stellate cells undergoing activation and acquiring a fibrogenic phenotype, while the level of TβRI increases or remains unchanged [17]. There is evidence suggesting that the weakening of TβRII expression is transient; e.g., during liver regeneration after its experimental damage with carbon tetrachloride, the expression of mRNA for TGF-β1 in non-parenchymal liver cells is most strongly expressed already 48 h after the damage; however, this is accompanied by a transient weakening of TβRII expression for a period of 72 h, which may be the cause of resistance to the antiproliferative and thus anti-regenerative effect of TGF-β [4]. TβRII, which is characterized in many cell types by a shorter half-life compared to TβRI and the described quantitative changes (mediated by TGF-β) on the cell surface, seems to play a key role in regulating the cell’s response to the discussed growth factor. The role of TβRII in modulating the cellular response to TGF-β was tested many years ago in a mink model, in which lung epithelial cells were transfected with dominant-negative TβRII. Transfected cells showed complete insensitivity to the growth-inhibiting effect of TGF-β; however, induction of PAI-1 and fibronectin synthesis was observed in them. Based on the above observations, the authors of the study concluded that TβRI and TβRII mediate the modulation of different cellular responses independently. Further studies did not confirm the above thesis [17].

## 5. Liver Fibrosis

In liver diseases of various etiologies (e.g., chronic viral hepatitis, alcoholic liver disease, and autoimmune hepatitis), there is a disturbance in the balance between the process of connective tissue production (fibrogenesis) and its degradation (fibrolysis). Fibrogenesis is a dynamic process characterized by quantitative and qualitative changes in the composition and distribution of the extracellular matrix [25]. Liver fibrosis is characterized by the excessive accumulation of extracellular matrix proteins, caused by both increased synthesis and decreased degradation of ECM proteins [26].

Ito cells (synonyms: hepatic stellate cells, perisinusoidal liver cells, lipocytes, and lipid-storing cells) are located in the perisinusoidal space (Disse’s space), and their main morphological feature is the presence of numerous lipid vacuoles (occupying about 20% of the cell volume) and long processes that often wrap around the wall of the liver sinusoid. Under normal conditions, stellate cells contain 70–80% of the body’s vitamin A stores, playing a key role in the metabolism of this compound. Under the influence of compounds released from damaged hepatocytes, as well as from Kupffer cells, platelets, white blood cells, and endothelial cells, there is a transformation (activation) of quiescent stellate cells into cells with a myofibroblast phenotype. PDGF has been shown to stimulate the proliferation of Ito cells, while TGF-β increases the synthesis of glycosaminoglycans by these cells [4,19].

The activation of Ito cells is manifested by their enlargement, increased proliferation, loss of lipid vacuoles (loss of vitamin A), and the appearance of α-smooth muscle actin (α-SMA) [4]. The presence of α-actin, typical of smooth myocytes, and the presence of receptors for many vasoactive substances allow stellate cells to contract and modulate blood flow in the liver sinusoids [25]. The activation of stellate cells is also accompanied by the proliferation of the granular endoplasmic reticulum, which is the cause of the abundant production of ECM proteins deposited in the Disse spaces and is the beginning of the fibrosis process, ultimately leading to liver cirrhosis. Fibrogenesis is a very efficient process; connective tissue scars in acute necrotic liver damage appear as early as 7–10 days after the disease. In the initial stages of fibrosis, the connective tissue matrix is deposited in the perisinusoidal spaces, which become enlarged as a result. Perisinusoidal fibrosis causes the wall of the sinusoids to undergo capillarization, i.e., it acquires a basement membrane and loses fenestration [22]. ECM proteins deposited in the perisinusoidal space make metabolic exchange between blood and hepatocytes difficult and lead to their damage [19]. The most important of these proteins are collagens (types I, III, IV, and VI), fibronectin, laminin, entactin, tenascin, and undulin, as well as glycoproteins and proteoglycans, which are a group of protein-polysaccharide compounds [4,22,25].

In a healthy liver, type I and III collagen constitute about 80% of total collagen. During fibrosis, type I collagen predominates; it is degraded by collagenase derived from stellate cells. Type III collagen, on the other hand, is degraded by enzymes derived from inflammatory cells, i.e., trypsin and elastase in neutrophils [19].

The expression of mRNA for collagen I in the liver is a good indicator of fibrogenic activity. Increased mRNA levels for TGF-β have been shown in adipocytes, myofibroblasts, and liver fibroblasts in rats with carbon tetrachloride-induced fibrosis. mRNA for TGF-β was also detected in the cells of liver inflammatory infiltrates. The serum concentration of N-terminal procollagen III peptides can also be a marker of liver fibrosis [5,19].

The effect of TGF-β on collagen production depends partly on the presence of vitamin A. The loss of vitamin A stores by sinusoid cells leads to their transformation into collagen-producing cells similar to myofibroblasts. However, even in cells enriched in vitamin A, collagen production is increased by TGF-β [19].

As mentioned earlier, stellate cells produce not only most of the extracellular matrix components of the liver lobule but also enzymes involved in their degradation [25].

Among the factors counteracting fibrosis, a special place is occupied by collagenase, an enzyme that exhibits the properties of selective digestion of type I collagen [19]. TGF-β1 inhibits the synthesis of proteolytic enzymes involved in the degradation of matrix proteins and metalloproteases (e.g., collagenase) and clearly stimulates tissue inhibitors of metalloproteases (TIMPs). Additionally, TGF-β1 induces the expression of plasminogen activator inhibitor (PAI)-1. This leads to a decrease in the conversion of plasminogen to plasmin, a protease that directly degrades matrix components and activates metalloproteases [4,17].

TGF-β was mainly localized extracellularly in the portal and periportal spaces in livers with chronic active inflammation and active cirrhosis. On the other hand, TGF-β was not detected in end-stage cirrhosis without signs of disease activity. This indicates the participation of this factor in the development and progression of liver fibrosis [19].

The influence on both the synthesis and degradation of the ECM confirms the role of TGF-β1 as the main autocrine and paracrine regulator of liver fibrosis, both in experimental conditions and as a result of viral hepatitis. This is confirmed by the correlation of TGF-β1 mRNA expression with ECM protein deposition, as well as the demonstration of a correlation between TGF-β1 concentration, determined in the urine of patients with liver cirrhosis, and the concentration of the amino-terminal propeptide of procollagen type III (PIIINP) in serum. Also, in patients with chronic hepatitis type C, the serum concentration of TGF-β1 shows a strong correlation with fibrosis analyzed histologically. In Ito cell culture, TGF-β1 stimulates collagen synthesis by up to 235% even at a concentration of 10 ng/mL. The mean plasma concentration of TGF-β1 in patients with chronic liver diseases exceeded 30 ng/mL, which may be a strong stimulus stimulating liver fibrosis [4].

TGF-β2 and TGF-β3, although to a lesser extent, also stimulate non-parenchymal cells to produce procollagen I and III mRNA. In addition to the direct effect, TGF-β enhances the mitogenic effect of PDGF, stimulating the proliferation of Ito cells. It has been shown that the increased expression of TGF-β3 associated with fibrosis is inhibited by exogenous prostaglandins E PGE1 and PGE2, which may explain the mechanism of therapeutic action of these prostaglandins. Studies on human fibroblast cultures suggest that TGF-β-stimulated PGE2 production may play a role in the autoregulatory mechanism limiting the effect of TGF-β on ECM protein synthesis [4].

The possibility of using a soluble form of the receptor for TGF-β (sTGF-R) raises great hopes, which, competing with TβRII for binding TGF-β, inhibits the fibrogenic effect of the discussed growth factor. The effectiveness of the antifibrogenic effect of the chimeric soluble receptor consisting of the extracellular domain of rabbit TβRII, fused to human IgG, was tested in an animal model. The extracellular domain of rabbit TβRII, which binds TGF-β, is 83% homologous to the rat TβRII domain. Male Sprague Dawley rats were given a slow intravenous infusion of the above-described sTGF-βR at the time of bile duct ligation (day zero, group I) or 4 days after the procedure (group II). In parallel, rats in the control group were given an infusion of IgG alone. It turned out that the infusion of the soluble receptor blocks the expression of mRNA for collagen type I in vivo. In animals that received the soluble receptor on day zero, the expression of the gene encoding collagen type I in stellate cells on day 4 after bile duct ligation was maintained at 26% compared to the control group. A similar relationship was observed in group II, where the expression of mRNA for collagen type I on day 8 after the procedure was 35% compared to the control group. In addition, the infusion of the soluble receptor inhibits the proliferation and activation of stellate cells. In both groups I and II, sTGF-βR infusion reduced the expression of smooth muscle actin in stellate cells [21].

The inhibition of TGF-βs’ fibrogenic effects by the soluble receptor sTGF-βR opens up the possibility of its therapeutic use in individuals with chronic liver diseases, which are typically accompanied by an intensified process of fibrosis [21].

The aforementioned studies clearly attribute a key role to TGF-β in liver fibrosis. However, it is important to remember that TGF-β is not the only cytokine that influences the process of liver fibrosis. Numerous other factors also appear to play a significant role in modulating liver regeneration. One such factor is hepatocyte growth factor (HGF), which interacts with cells via receptors with tyrosine kinase activity. HGF can exert effects opposite to those of TGF-β and is a potent proliferative factor for hepatocytes. Moreover, in a rat model used to study liver regeneration, elevated serum levels of HGF (derived from transfected muscle cells) suppressed fibrosis and stimulated hepatocyte proliferation. Additionally, in cells treated with both HGF and TGF-β, HGF functionally dominates and blocks the effects of TGF-β. Therefore, changes in HGF activity may contribute to TGF-β-mediated liver fibrosis [13].

## 6. Autoimmune Hepatitis

### 6.1. The Liver as an Immunologically Privileged Organ

The liver, like the gonads and eyes, is an immunologically privileged organ [27]. This is because a large number of antigens from food or commensal bacteria reach the liver through the portal vein. For example, the concentration of endotoxins in portal vein blood is much higher than in peripheral blood, suggesting that the liver somehow “clears” the blood of antigens and endotoxins [28]. Additionally, as the main metabolic organ, the liver produces many neo-antigens, which increases the risk of immune activation within the liver and other organs. For these reasons, there are many mechanisms of immunological self-tolerance in the liver.

Dendritic cells, liver sinusoidal endothelial cells, Kupffer cells, hepatic stellate cells, and regulatory T cells (Tregs) are involved in this process. These cells secrete the cytokines IL-10 and TGF-β, as well as co-stimulators involved in programmed cell death (PD-L1). Dendritic cells, Kupffer cells, and liver sinusoidal endothelial cells produce the anti-inflammatory IL-10. Increased synthesis of PD-L1 in liver sinusoidal endothelial cells, stellate cells, and hepatocytes causes inactivation of CD8+ lymphocytes during antigen presentation. Stellate cells, through the storage and release of TGF-β and vitamin A, can contribute to the conversion of T cells into Tregs. Tregs mediate active tolerance by suppressing T cell responses via IL-10, TGF-β, and CTLA-4. Disruption of these mechanisms can lead to autoimmune liver inflammation [29].

### 6.2. Autoimmune Hepatitis

Autoimmune hepatitis (AIH) is a disease that is becoming increasingly prevalent. Compared to rates before 2000, the incidence and prevalence of AIH have increased 3.1-fold and 2.8-fold, respectively. The disease is associated with a two-fold increased risk of death and can contribute to acute and chronic liver failure. The global overall incidence rate is 15.65 cases per 100,000 inhabitants. Moreover, AIH incidence and prevalence were higher in high HDI populations (>0.92), in women, in adults over 65 years of age, in the North American population (compared to Europe, Asia, and Oceania), and at high latitudes (>45°) [30].

There are two types of AIH: type 1 is characterized by antinuclear antibodies (ANA) and/or anti-smooth muscle antibodies (SMA); type 2 is characterized by anti-liver-kidney microsome antibodies (anti-LKM 1). The etiology of AIH is still unknown, but environmental and genetic factors are involved. It is characterized by elevated serum transaminase activity, positive organ-specific and non-organ-specific autoantibodies, and elevated IgG levels. The histological picture shows interface hepatitis—a dense infiltrate of mononuclear cells attacking the surrounding parenchyma and including T and B lymphocytes, macrophages, and plasma cells [3]. TGF-β inhibits the development of Th1 cells and promotes the development of Th2 cells. Importantly, the transition of Th0 cells to Th1/Th2 cells is crucial for generating effective immunity and avoiding autoimmunity [31]. Overexpression of TGF-β is observed in AIH and correlates with its activity. It is probably a regulatory mechanism towards restoring homeostasis, at the cost of increased fibrogenesis and ultimately leading to liver cirrhosis [32].

### 6.3. Genetic Background of AIH

In the case of impaired TGF-β signaling in transgenic mice with overexpression of a dominant-negative (inhibitory) TβRII receptor in T cells with induced AIH (immunized with syngeneic liver homogenate in adjuvant), there is increased leukocyte infiltration into the liver (mainly T cells), hepatocellular necrosis, increased IFN-γ production, and decreased IL-4 production, which in turn promotes a cellular response [31,32]. In wild-type mice after immunization and in mice with overexpression of the dominant-negative TβRII receptor that were not immunized, the above-mentioned relationships were not observed [31]. This indicates the role of TGF-β in suppressing the previously developed immune response and the response to liver inflammation.

Similar experiments were performed in BULB/c mice homozygous for the knocked-out TGF-β1 allele (TGF-β1^−/−^), with the difference that these mice developed spontaneous AIH, in contrast to mice with overexpression of the dominant-negative TβRII receptor, in which AIH requires prior immunization. The phenotype of a given TGF-β1^−/−^ mouse depends largely on the genetic background. In C57BL/6-TGF-β1^−/−^ mice, all fetuses die in utero due to impaired blood vessel development in the yolk sac. In contrast, 129/CF-1-TGF-β1^−/−^ and BALB/c-TGF-β1^−/−^ mice are born alive, but BALB/c-TGF-β1^−/−^ mice show inflammatory changes in the liver, while 129/CF-1-TGF-β1^−/−^ mice do not [32]. This suggests the existence of different genetic predispositions that increase the risk of AIH.

In humans, various single nucleotide polymorphisms (SNPs) of the HLA (human leukocyte antigen), IL-10, and TGF-β genes have been identified that are more common in people with AIH [33,34]. HLA molecules enable antigen presentation and T cell activation, which is why they are important in the pathogenesis of AIH. Several different polymorphisms have been identified that are associated with an increased susceptibility to AIH type 1 or type 2, and different polymorphisms have been identified in different populations. For example, the presence of HLA DR3 (DRB10301) and DR4 (DRB10401) is associated with a higher incidence of AIH type 1, and HLA DR7 (DRB10701) and DR3 (DRB10301) with a higher incidence of AIH type 2 [35].

AIH type 1 can be divided into a pediatric form (pediatric autoimmune hepatitis, PAH) and an adult form (adult autoimmune hepatitis, AAH). They differ in the severity of the disease; the adult form is associated with less inflammation, less fibrosis, and a generally better course. Despite more immunosuppression, pediatric patients are more likely to progress to cirrhosis and require liver transplantation more often. These differences may be due to differences in TGF-β gene polymorphism. Two main SNP loci have been identified for TGF-β-codon 10 and codon 25. The 25GG genotype and the 10CC genotype are considered “high producer” genotypes. The 25GG genotype is associated with a more aggressive disease course than 10CC. For this reason, the 25GG genotype is more common in PAH. The 25CC “low producer” genotype and the 10CC genotype are more common in AAH. It has been shown that the co-occurrence of 25CC and 10CC eliminates the “high producer” effect of 10CC; hence, patients with this genotype and AAH have less inflammation, less fibrosis, and a generally better course [33].

### 6.4. The Importance of Regulatory T Cells

Regulatory T cells (Tregs) are extremely important in immunological self-tolerance, and their impaired function is observed in AIH. These cells have phenotypic and functional plasticity, which allows them to transform into effector T cells (Teff), promoting an inflammatory response. The predominance of Teff cells over Tregs leads to increased hepatocyte cytotoxicity and increased secretion of the pro-inflammatory cytokines IFN-γ and IL-17. Impaired Treg function can result from reduced cell number, impaired function, increased plasticity, and reduced metabolism. Since low-dose IL-2 can reverse the conversion of Tregs to Teffs and rebuild the Treg pool, low-dose IL-2 therapy is being considered for the treatment of AIH [36].

In mice, a relative increase in Treg frequency in the liver occurs around 1–2 weeks after birth, which appears to be driven by colonization of the intestine with commensal bacteria and is mediated by a pathway that requires TGFβ-1 and MyD88. MyD88 is an adaptor molecule that mediates the signaling of all TLR receptors that recognize bacterial patterns. It has been observed that in TGF-β1^−/−^ knockout mice, Myd88^−/−^ mice, or mice treated with antibiotics, the increase in Tregs after birth is smaller. In a model of Treg induction from colon colonic bacteria with Clostridium, it was shown that intestinal epithelial cells secrete more TGF-β1 and increase the expression of matrix metalloproteinases (MMP2, MMP9, and MMP13), which hydrolyze latent TGF-β, converting it to its active form. It is possible that the TLR/MyD88 response to microbial colonization in the intestine increases TGF-β production, leading to an increase in Treg numbers in the liver [36]. These relationships could explain the increased incidence of AIH in patients with frequent antibiotic therapy and intestinal dysbiosis [31] (Figure 3).

## 7. Chronic Hepatitis C

### 7.1. Hepatitis C Virus

Hepatitis C virus (HCV) is a major public health concern, affecting an estimated 58 million people worldwide. The majority of patients (80% to 85%) who develop acute infections progress to chronic disease. Consequences of chronic infection include liver cirrhosis, portal hypertension, liver decompensation with encephalopathy, and hepatocellular carcinoma [37].

In developed countries, the prevalence of HCV is typically 1–2%. HCV transmission requires contact between infectious virions and susceptible cells that allow replication. Hepatitis C virus RNA can be detected (including in serum and plasma), saliva, tears, and cerebrospinal fluid. Available data suggest that HCV can be transmitted through sexual intercourse, but this is rare. In the majority of patients with HCV in the United States and Europe, infection occurs as a result of intravenous drug use or unsafe medical practices in resource-limited areas of the world [37].

The distinction between patients in different populations who are in the acute phase of HCV infection and those with chronic infection is based on the determination of the immunological outcome of such patients. The immunological outcome of hepatitis C virus infection depends on a subtle balance between a strong immune response that can clear the infection but with a risk of non-specific inflammation or a lower inflammatory response that leads to chronic infection [38].

In patients with chronic HCV infection, there is exhaustion and impairment of the cytotoxic function of HCV-specific T cells and NK cells. This is primarily seen as an increased production of IL-10, probably due to increased levels and function of anti-inflammatory regulatory lymphocytes (Tregs). The main immune system factors during chronic HCV infection are characterized by decreased cytotoxic function and increased inhibitory function. This may be a way to reduce intrahepatic and systemic inflammation.

TGF-β is involved in the progression of chronic hepatitis C by controlling viral replication and mediating inflammation-related responses. TGF-β is essential for the induction and function of iTreg. The effect of HCV on T cell activity has been studied: HCV-infected hepatocytes co-cultured with CD4+ T cells increased the number of Tregs, and TGF-β was involved in Treg induction [39].

### 7.2. The TGF-β/SMAD Signaling Pathway in HCV Infection

In liver tissue, TGF-β isoforms are produced and secreted not only by non-parenchymal cells but also by hepatocytes. TGF-β isoform expression is promoted by HCV infection in the cultured hepatocytes and livers of HCV-infected patients.

While TGF-β plays a critical role in the progression of liver diseases, including initial liver injury, fibrosis, cirrhosis, and hepatocellular carcinoma, TGF-β has been shown to inhibit HCV RNA replication and protein expression in a subgenomic HCV replicon system in a TGF-β/Smad signaling pathway-dependent manner [40].

HCV can alter Smad activity through a variety of molecular mechanisms. HCV-induced fibrosis and cirrhosis are associated with increased local TGF-β and Smad-3/Smad-4 signaling. The HCV Core protein increases TGF-β transcription, which likely contributes to a positive feedback loop reinforcing fibrosis during chronic HCV infection.

However, another study showed that during HCV infection, both the Core and NS3 proteins directly bind Smad-3 and attenuate the activity of TGF-β and Smad-responsive reporters (Figure 4). Additionally, expression of both HCV Core and NS3 proteins inhibited TGF-β-induced apoptosis in vitro. NS3 can also directly bind and activate TβRI, acting as a TβRI ligand to promote liver fibrosis. Conversely, the HCV NS5A protein has been reported to directly interact with TβRI, inhibiting the phosphorylation of SMAD2, which has antifibrogenic effects, in contrast to Smad-3, which promotes fibrosis. On the other hand, the same study showed that NS5A inhibits the formation of the Smad-3/Smad-4 complex independently of Smad protein levels (Figure 4). Indeed, both Smad-2/Smad-3 and Smad-3/Smad-4 complexes inhibited HCV replication in vitro in a replicon model. Furthermore, TGF-beta isoforms, TGF-β1 and TGF-β2, inhibited HCV infection at three levels: virus entry, replication, and spread. Thus, modulation of Smad activity may adversely affect HCV infection initiation but may lead to chronic liver infection [41].

### 7.3. The Effect of HCV Infection on TGF-β Isoform Levels

The antiviral activity and underlying mechanisms of TGF-β isoforms in suppressing HCV propagation have been studied. All three TGF-β isoforms have been shown to inhibit HCV spread by interrupting several distinct steps throughout the HCV life cycle, including viral entry and intracellular replication, in both TGF-β/Smad-dependent and TGF-β/Smad-independent signaling pathways. The latter pathway is independent of the canonical TGF-β signaling pathway and therefore exhibits additional anti-HCV activity [41].

New studies demonstrate that serum TGF-β1 levels are significantly decreased after treatment with pegylated interferon alfa and ribavirin in patients with chronic hepatitis C. Additionally, serum TGF-β1 levels are mainly decreased in patients with chronic hepatitis C who achieve sustained virologic response [42].

New studies also show that serum TGF-β1 and Il-17 levels are associated with the degree of liver inflammation and fibrosis stages. TGF-β1 and IL-17 may be promising serum biomarkers for monitoring the progression of liver inflammation and fibrosis associated with chronic HCV infection. Therefore, they could be used in the future as targets for antifibrotic therapy for chronic HCV to mitigate disease progression [43].

## 8. Non-Alcoholic Fatty Liver Disease

### 8.1. Essence of the Disease

Non-alcoholic fatty liver disease (NAFLD) has become a common liver disorder worldwide, affecting 25% of the general population and up to 80% of obese individuals. While the exact process of NAFLD development is not fully understood, it is often explained by the classical “multiple hit” pathogenesis theory, which proposes that fat accumulation leads to liver steatosis, triggering a series of insults such as adipokine secretion, inflammation, lipotoxicity, and impaired glucose and lipid metabolism, eventually progressing to non-alcoholic steatohepatitis (NASH) and liver cirrhosis [44,45].

NAFLD is a spectrum of liver conditions that encompasses various stages of disease progression, ranging from simple fatty change (FL), the least clinically significant form, to the intermediate and usually progressive stage, non-alcoholic steatohepatitis (NASH), to the most severe stage, liver cirrhosis. The diagnosis of this disease entity is typically based on imaging findings, provided that other liver diseases are excluded. Liver biopsy is necessary to assess the disease stage, including inflammation, ballooning, and fibrosis, although certain limitations question its utility in clinical practice [46].

Therefore, the search for predictors and useful markers of NAFLD is a natural course of action. A meta-analysis conducted for 16 inflammatory cytokines, including transforming growth factor β, found no significant associations of this cytokine with NAFLD in the general population but did find associations in subgroup analyses. For example, in the Asian population, this disease entity was significantly associated with TGF-β, TGF-α, and CRP. Similarly, for the age group 18–60 years, an association was shown with the aforementioned cytokines and ICAM-1. When studies were stratified by BMI ≥ 30, NAFLD was found to be associated with CRP and TGF-β. This group also showed significantly reduced heterogeneity (I^2^ = 0.0%, *p* = 0.934) due to the sample source and diagnostic methods for NAFLD related to TGF-β. Additionally, the results showed a statistically significant association of TGF-β with NAFLD for studies performed on serum samples.

The low level of association of TGF-β with NAFLD in the general population but high in subgroups can be explained by the limited number of studies included in the meta-analysis and the multiple features of inflammatory cytokines involved in NAFLD progression. These results suggest that the true relationship between inflammatory cytokines, including TGF-β, and NAFLD may have been masked or diluted when analyzing the entire population [47].

### 8.2. Plasma and Serum TGF-β Levels

TGF-β1 differs from most growth factors in that it is usually produced and secreted in a biologically inactive form and must be activated before exerting its biological effects on target cells. In its inactive form, it has a long half-life in plasma (>100 min). The detection of increased amounts of TGF-β1 in the blood of patients with fatty livers suggests that there is probably an overexpression of this cytokine in this organ [46].

In a study, TGF-β was measured in the plasma of 37 obese patients using ELISA. A positive correlation was found with abnormal ultrasound findings, particularly considering the presence and degree of steatosis. Moreover, in the same study, genotyping of the TGFB1 promoter region polymorphism, specifically TGFB1 G-800A (rs1800468) and TGFB1 C-509T (rs1800469), was performed in 75 obese and 45 eutrophic patients using polymerase chain reaction-restriction fragment length polymorphism (PCR-RFLP). It was shown that the GT haplotype, which carries the C-509T allele, associated with higher TGF-β1 production, is associated with the same liver disorder markers as elevated plasma TGF-β1 levels, suggesting a genetic basis for this relationship and indicating that increased TGF-β1 production caused by this genetic polymorphism may increase the risk of liver complications in obese patients. Such a positive, constant correlation at both the molecular and patient plasma levels may indicate the usefulness of transforming growth factor β as a potential biomarker for this type of liver disorder in obese individuals [48].

Another study focused on elevated serum TGF-β3 levels as a predictive marker for NAFLD development in healthy individuals. The study included 1322 individuals without diseases that could affect the final conclusions of the current study, and then the development of NAFLD in this population was assessed after 4 years, in 2013. In individuals who developed the disease (25.3% of the study population, 334/1322), TGF-β3 levels were significantly elevated in 2009, highlighting the role of the cytokine as a potential predictor of NAFLD development [49].

The presence of TGF-β in plasma and its correlation with NAFLD were also investigated in pediatric patients. The study included 72 overweight adolescents aged 10–19 years, of whom 36 had fatty liver. In multivariate analysis, plasma TGF-β levels were associated with the presence of fatty liver, independent of other variables. Discriminant analysis confirmed that TGF-β levels can be an indicator of fatty liver cases but do not reflect the severity of the cases in the study group [50].

### 8.3. Molecular Background of Non-Alcoholic Fatty Liver Disease

Fatty liver with insulin resistance is associated with oxidative stress, lipotoxicity, adipokine secretion by adipocytes, endotoxins (lipopolysaccharides) released by the gut microbiota, and endoplasmic reticulum stress. These factors contribute to the progression of NAFLD to NASH, liver fibrosis, and cancer. Transforming growth factor TGF-β is a cytokine involved in pathological processes such as liver fibrogenesis and carcinogenesis [51].

The basis of fibrosis is the process of fibrogenic activation and transdifferentiation of hepatic stellate cells (HSCs) into myofibroblasts (MFBs). It can be summarized in a three-stage cascade model. The first stage, initiated by the pre-inflammatory phase, involves the direct paracrine activation of HSCs by dead hepatocytes, resulting in the release of activating cytokines. In the next inflammatory phase, pre-activated HSCs are further stimulated paracrinally by invasive leukocytes, platelets, Kupffer cells, sinusoidal endothelial cells, and hepatocytes, leading to their transdifferentiation into MFBs. In the third stage, the cross-linking phase, MFBs secrete stimulating cytokines and interacting matrix components. Some of these cytokines can stimulate MFBs in an autocrine manner and unactivated HSCs in a paracrine manner. TGF-β, secreted as large molecules, is bound by stellate cells (HSCs/MFBs), sinusoidal endothelial cells, and Kupffer cells, and is also released by damaged platelets and hepatocytes. This cytokine not only initiates the activation of HSCs in MFBs but also increases the expression of matrix genes, decreases their degradation by limiting the activity of matrix metalloproteinases and increasing specific inhibitors (tissue metalloproteinase inhibitors, TIMPs), induces hepatocyte apoptosis, and inhibits (together with activin A) liver cell proliferation [52]. In chronic liver diseases, MFBs remain active, proliferate, and then migrate while continuing to deposit extracellular matrix (ECM) that replaces damaged liver parenchyma. 

Additionally, recent studies suggest the possibility of interaction between Sonic Hedgehog (Shh) proteins and TGF-β1 in liver inflammatory processes. Shh secretion can stimulate TGF-β1 activity, leading to HSC activation and contributing to NASH progression in humans [51]. 

Antagonism of TGF-β action or inhibition of its intracellular Smad signaling cascade using specific inhibitors leads to a significant slowdown in the activation of hepatic stellate cells (HSCs) and thus to a lasting antifibrotic effect [9]. The TGF-β/Smad3 signaling pathway in the liver is also a well-known key factor in advanced non-alcoholic fatty liver disease through the progression of liver fibrosis. Smads, as signal transducers possessing MH1, MH2, and linker domains, transmit signals from membrane receptors for members of the TGF-β superfamily to the cell nucleus [53]. TGF-β binds to type II membrane receptors, leading to the activation of type I receptors (TβRI). Upon activation of TβRI, Smad3 is phosphorylated at the C-terminus, resulting in the formation of an isoform known as pSmad3C. TGF-β- and pSmad3C-dependent signals interfere with cell cycle progression by stimulating the expression of p15INK4B and p21CIP1 genes while simultaneously inhibiting c-Myc gene expression.

TGF-β also triggers the activation of Smad-independent signaling pathways, which include TAK1 (whose deletion causes hepatocellular carcinoma (HCC)) and JNK. In addition to regulating transcriptional responses, phosphorylation of the Smad linker protein in the cytoplasm plays an important role in integrating JNK signaling with the TGF-β pathway. Phosphorylation of the Smad3 linker protein allows Smad translocation to the nucleus, where further consequences of JNK signaling occur. JNK activates the activator protein (AP)-1, which stimulates cyclin D expression and initiates the transition from the G0 to G1 phase of the cell cycle [51]. It can be hypothesized that JNK signaling is involved in various oncogenic pathways leading to HCC development [54]. Increased JNK activity has been found in the livers of patients with obesity and NAFLD. In studies, mice lacking JNK1, those lacking JNK2, and those heterozygous for JNK1 loss of function, as well as mice lacking both JNK1 and JNK2 in their livers, were significantly protected from liver fat accumulation during diet-induced obesity experiments. These observations suggest that JNK activity inhibition may prevent steatosis, acting both directly and indirectly [51]. The importance of TGF-β/Smad3 signaling and cross-talking pathways is also highlighted by the development of drugs targeting this mechanism. For example, SP-1154, a novel synthetic derivative of verbenone, inhibits the complexation of TGF-βR1 and TGF-βR2, which, in turn, inhibits Smad3 phosphorylation. This is because the interaction between TGF-β1 and the TGF-βR1 and TGF-βR2 complex is a key factor initiating the TGF-β/Smad3 signaling pathway.

In preclinical studies, the use of TGF-β-neutralizing monoclonal antibodies has also been shown to inhibit this signaling pathway. The aforementioned TGF-β inhibitor has been shown to reduce obesity and liver steatosis in mice fed a high-fat diet, which provides hope for patients struggling with fatty livers [55] (Figure 5).

## 9. Summary

The above arguments clearly indicate the pleiotropic effects of transforming growth factor-β. It is of utmost importance in both physiological and pathological states of the liver. The relationship between TGF-β activity and other biologically active molecules, as well as tissue and organ specificities, make it difficult to unequivocally classify this cytokine.

However, suppression of the immune response, participation in the scarring process, and the broad regulation of cell growth and differentiation are constant and undeniable physiological functions of TGF-β. Additionally, TGF-β inhibits DNA synthesis in many cells, impairing their growth and thus inhibiting the proliferation of individual cell types by blocking their transition from the G1 to S phase of the cell cycle. This factor also plays an important role in the inflammatory response, increases the body’s susceptibility to infections, and inhibits the proliferation of B and T lymphocytes. TGF-β at a concentration exceeding the growth inhibitory concentration induces apoptosis in many cells, including liver cells. Nevertheless, this cytokine is physiologically secreted by normal non-parenchymal liver cells, being a strong inhibitor of hepatocyte growth. Many pathological liver conditions, such as liver fibrosis, chronic hepatitis C, autoimmune hepatitis, and non-alcoholic fatty liver disease, show a strong correlation with TGF-β-related signaling.

## Figures and Tables

**Figure 1 biomedicines-12-00925-f001:**
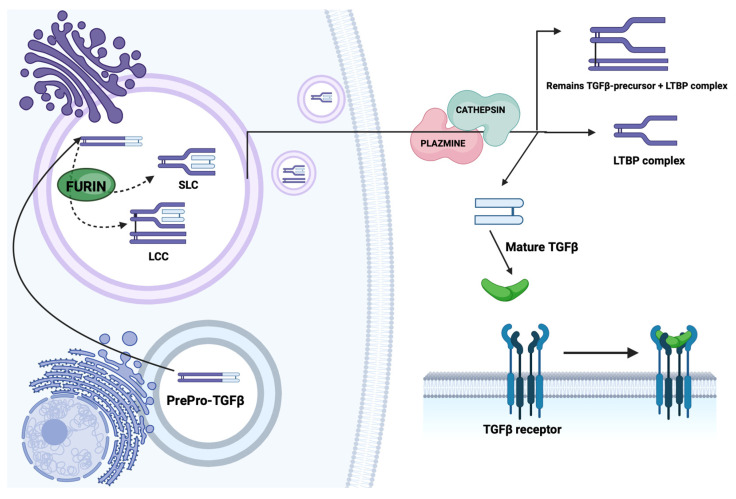
Precursor synthesis, fission in the AG, exocytosis of SLC and LCC, hydrolysis, and maturation of TGF-β.

**Figure 2 biomedicines-12-00925-f002:**
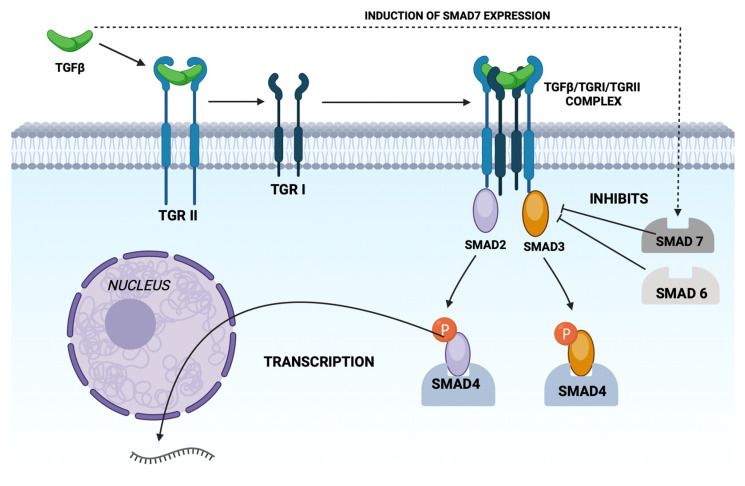
Integration of mature TGF-β with the membrane receptor; creation of the membrane receptor complex; and induction of the signal to the nucleus via the SMAD family. The dotted line indicates positive feedback induction of the SMAD7 inhibitor.

**Figure 3 biomedicines-12-00925-f003:**
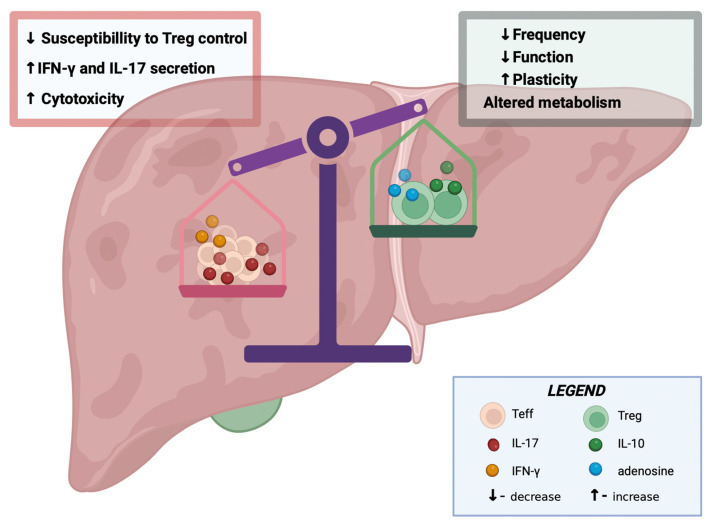
Treg impairment in AIH. The imbalance of Treg and Teff cells fuels tissue damage in AIH by allowing overactive T cells to attack liver cells with increased cytotoxicity and secretion of proinflammatory cytokines like IFN-γ and IL-17. The impairment of Treg cells can stem from various factors such as reduced frequency, defective function, an increased tendency to acquire effector cell features (plasticity), and altered metabolism that limit their ability to produce certain molecules like adenosine and IL-10. Consequently, immune cells from AIH patients are less responsive to regulatory control by Tregs.

**Figure 4 biomedicines-12-00925-f004:**
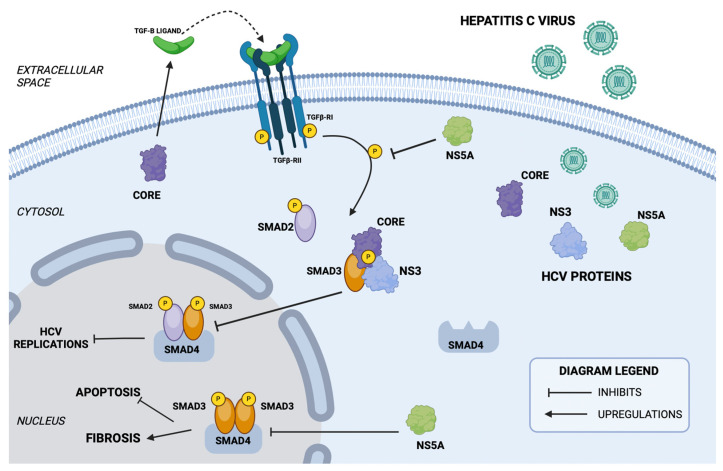
HCV modulates Smad signaling at different levels. HCV core protein increases TGF-β transcription and inhibits Smad-2/Smad-3 complex formation and signaling through direct interaction with Smad-3. NS3 also inhibits Smad-3 complex formation. NS5A inhibits R-Smad phosphorylation, in effect reducing TGF-β-induced apoptosis [41].

**Figure 5 biomedicines-12-00925-f005:**
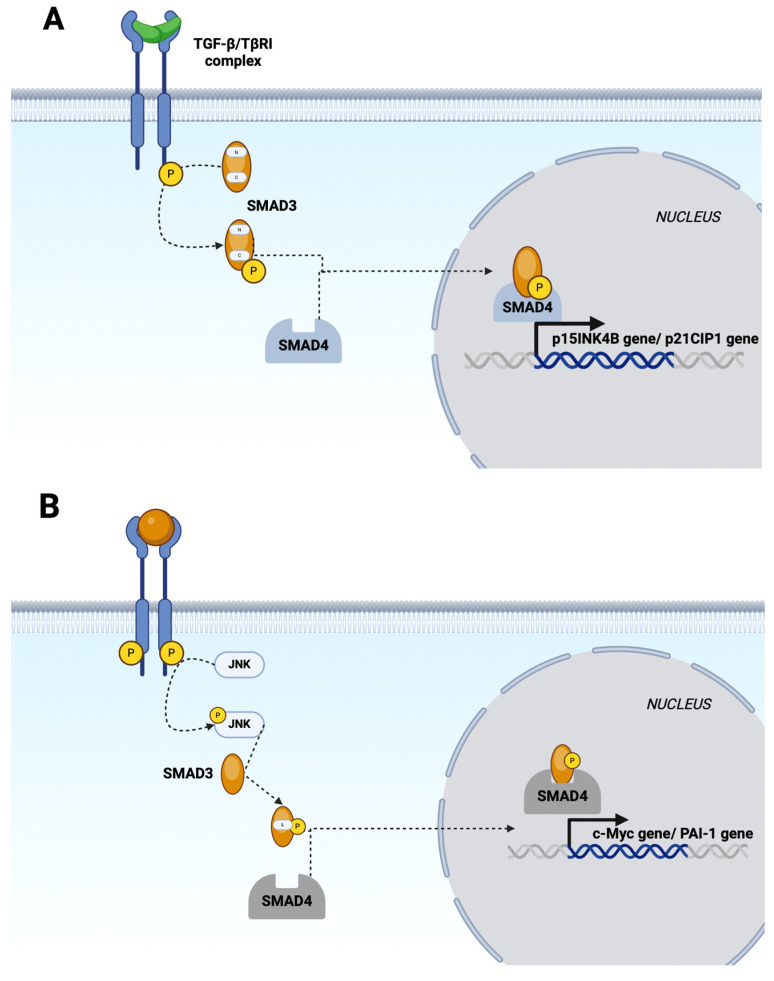
(**A**) As a result of activation of the type I transforming growth factor-β (TGF-β) receptor (TβRI), the serine residues at the C-terminus of the Smad-3 molecule are phosphorylated. The phosphorylated Smad-3C then translocates, together with Smad-4, to the cell nucleus, where it affects cell growth inhibition by stimulating (bold arrow) the expression of the p15INK4B and p21CIP1 genes. (**B**) Pro-inflammatory cytokines, such as tumor necrosis factor-α (TNF-α), activate the N-terminal c-Jun fragment kinase (JNK), causing phosphorylation of the Smad3 linker region. Phosphorylated at the Smad3 linker region (pSmad3L) translocates with the Smad4 complex into the cell nucleus, leading to the activation of c-Myc expression, which stimulates cell proliferation, and plasminogen activator inhibitor type 1 (PAI-1), which promotes cell invasion and migration.

## Data Availability

Not applicable.

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
