# Peer review of "Pleiotropic Action of TGF-Beta in Physiological and Pathological Liver Conditions"

_biomedicines, 2024, doi:10.3390/biomedicines12040925_

Round 1
Reviewer 1 Report
Comments and Suggestions for Authors
The manuscript of Braczkowski et al. deals with a review article on the pleiotropic effects of TGF-β in physiological and pathological conditions of the liver, with particular emphasis on its role in immune suppression, wound healing, regulation of cell growth and differentiation, and liver cell apoptosis. TGF-β plays a constant role in immune suppression, wound healing, and regulation of cell growth and differentiation. In concentrations exceeding the norm, it can induce apoptosis of liver cells. Increased TGF-β levels are observed in many liver diseases, such as fibrosis, inflammation, and steatosis. Authors concluded that TGF-β concentration may be a potential diagnostic and prognostic marker in liver diseases. From here, this review could be suitable for publication after addressing some concerns:
1) What is known about participation of TGF-β in human hepatocarcinoma, looking its role in cell proliferation and modulation of the immune system?
2) In the liver, TGF-α can induce changes opposite that TGF-β. What is known about a possible reciprocal regulation between these growth factors?
3) There are many abbreviations throughout the manuscript. Despite they are defined in the text, a possible list of abbreviations would be useful.
Author Response
Thank you very much for submitting your review. In response, I inform you that:
1. list of abbreviations is attached,
2. mutual regulation between tgf alfa and tgf beta was added in the text
3. information on the role of tgf beta in liver cancer has not been included because this issue is very extensive and due to the scope that our work has already reached, we have planned the role of tgf beta in liver cancer for a new next one article.

Reviewer 2 Report
Comments and Suggestions for Authors
The authors of the review are thanked for the quality of the work and the choice of the subject relating to the pleiotropic effects of TGF-β in physiological and pathological conditions of the liver, with particular emphasis on its role in immune suppression, wound healing, regulation of cell growth and differentiation, and liver cell apoptosis. Braczkowski and collaborators analyze 52 studies, comprising review articles, in vitro and in vivo studies, and meta-analyses. In conclusion, the authors indicate the pleiotropic effects of transforming growth factor - TGF-β, both in physiology and pathophysiology. TGF-β plays a constant role in immune suppression, wound healing, and regulation of cell growth and differentiation. In concentrations exceeding the norm, it can induce apoptosis of liver cells. Increased TGF-β levels are observed in many liver diseases, such as fibrosis, inflammation, and steatosis. TGF-β has been shown to play a key role in many physiological and pathological processes of the liver, and its concentration may be a potential diagnostic and prognostic marker in liver diseases.
The manuscript entitled “Pleiotropic Action of TGF-beta in Physiological and Pathological Liver Conditions” is scientifically valid and the manuscript is well-written, but can not be published in present form. My major concern is the lack of novelty. 52 references were cited in the manuscript. However, 9 of them were published over 10 years ago, and as many as 23 of them were published over 20 years ago. I realize that the manuscript is a review and is intended to present basic knowledge on a given topic, but it should also include the latest publications with the latest research. And there are too few of them in the manuscript.
If the authors were to resubmit this manuscript I would like to ask them to supplement their review with newer, most up-to-date research.
Author Response
Thank you very much for submitting your review. In response, I inform you that:
1. Several new different sources have been added. The role of TGF beta in specific disease syndromes is described mainly on the basis of new articles, but the basic information on the key functions of TGF beta actually comes from slightly older articles, but due to their very important role in understanding the structure and function of TGF beta, it is impossible to ignore them. in our article, therefore their inclusion seems fully justified to us and the option of supplementing the article with newer items rather than replacing older articles with newer ones.
Round 2
Reviewer 2 Report
Comments and Suggestions for Authors
Thank you for considering my comments. I have no further objections. The manuscript can be publish in present form.